# Surface Plasmon Nanolaser: Principle, Structure, Characteristics and Applications

**Litu Xu, Fang Li \*, Yahui Liu, Fuqiang Yao and Shuai Liu**

Hubei Key Laboratory of Optical Information and Pattern Recognition, School of Optoelectronics and Energy, Wuhan Institute of Technology, Wuhan 430205, China; 15871430579@163.com (L.X.); 15872431358@163.com (Y.L.); yfq15872268171@163.com (F.Y.); 15342251052@163.com (S.L.)

**\*** Correspondence: lifang@wit.edu.cn or lifang_wit@hotmail.com; Tel.: +86-027-8799-2024



**Featured Application:** **Plasmonic nanolasers use nanowires to integrate working materials and resonators. Surface plasmas can break through the optical diffraction limit; therefore, plasmonic nanolasers have the advantages of a small size, monochromaticity, good orientation, high efficiency, a low energy threshold and short response time. This kind of ultra-small laser has the potential for very broad application in a series of fields. For example, it can be used to automatically control switches in circuits; in chemical and biomedical engineering, it can be used in biosensors, microscopy, chemical identification and so on; and it can be used to improve the information storage capacity of computer disks when applied to chips.**

**Abstract:** Photonic devices are becoming more and more miniaturized and highly integrated with the advancement of micro-nano technology and the rapid development of integrated optics. Traditional semiconductor lasers have diffraction limit due to the feedback from the optical system, and their cavity length is more than half of the emission wavelength, so it is difficult to achieve miniaturization. Nanolasers based on surface plasmons can break through the diffraction limit and achieve deep sub-wavelength or even nano-scale laser emission. The improvement of modern nanomaterial preparation processes and the gradual maturity of micro-nano machining technology have also provided technical conditions for the development of sub-wavelength and nano-scale lasers. This paper describes the basic principles of surface plasmons and nano-resonators. The structure and characteristics of several kinds of plasmonic nanolasers are discussed. Finally, the paper looks forward to the application and development trend of nanolasers.

**Keywords:** plasmonic nanolasers; surface plasmons; nano-resonators; sub-wavelength

## 1. Introduction

As one of the greatest inventions of the 20th century, lasers have had an enormous impact on human production and life—as much as the discovery of fire. After more than 60 years of evolution, the laser is rapidly developing in the direction of smaller volume, higher power, higher efficiency, and faster modulation speed [1–6]. At the end of the 20th century, the rapid development of nano-materials and nano-technology had a profound influence on the fields of information and materials, providing conditions for the miniaturization of lasers. However, conventional semiconductor lasers are subject to the diffraction limit, and so their size cannot be less than half a wavelength, hindering the miniaturization of lasers [7–13].

Unlike conventional lasers, surface plasmon (SP)-based nanolasers can reduce the mode size and physical size of lasers to less than half a wavelength simultaneously, forming a nanoscale coherent light source far beyond the diffraction limit [14–19]. They can realize the unprecedented linear and

nonlinear enhancement of optical processes, nanoscale optical interactions, etc. [20–22]. The SP-based nanolaser is called the world's smallest laser and is the core component for achieving nanoscale optoelectronic integration. This ultra-small nanolaser has the potential for very broad application in a series of fields. It can be used in biosensors, microscopy, and laser surgery in chemical and biomedical engineering, and it is also possible to use nanolasers to identify chemicals. In the field of material processing, it can be used to fabricate and analyze metamaterials that are smaller than the wavelength of light emitted by current lasers. Supermaterials are used in super lenses that see a single virus or DNA molecule. At the same time, nanolasers are also widely used in optical computing, information storage and nano-analysis. Nanolasers can be used in circuits that can automatically regulate switching. Some lasers have been able to switch faster than 20 billion times per second. If the laser is integrated and mounted on the chip, the amount of computer disk information storage and the information storage capacity of future photonic computers can be improved, and the integrated development of information technology can be accelerated [23–25].

However, because surface plasmons need to be excited with metal, the presence of metal means that the laser has a high loss characteristic, resulting in a relatively short transmission distance of light [26–30]. In order to achieve a lower threshold and lower loss transmission, a variety of nanolaser structures have been designed, which can be generally classified into metal–dielectric–metal structure nanolasers, metal nanoparticles plasmonic nanolasers [31–34], nanowire surface plasmonic nanolasers, array nanowires plasmonic nanolasers, W-G mode plasmonic nanolasers, etc [35,36]. In addition, many research groups have applied excellent optical gain materials such as perovskite, $MoS_2$, graphene, and monolayer transition-metal dichalcogenides (TMDCs) to the research of nanolasers and made breakthroughs, effectively making up for the loss in the laser system [37–47]. This paper introduces plasmonic nanolasers that have been experimentally proven and have made breakthroughs in recent years. In particular, the plasmonic nanolaser based on metal halide perovskite nanowires that replace conventional semiconductor nanowires is highlighted.

## 2. Basic Principle of Surface Plasmons

The surface plasmon is a collective oscillation caused by the free electrons in the metal surface under the action of an external electromagnetic field. According to the form of expression, surface plasmons can be divided into propagation-type surface plasmon polaritons (SPPs) and non-propagating localized surface plasmons (LSPs) [48,49]. In this paper, the basic principles of surface plasmons are characterized by taking SPPs as an example. LSPs are generated by the interaction between the electrons in the particles and the photons of the incident light when the light is incident on the metal nanoparticles. SPPs are electromagnetic waves propagating along a metal surface generated by the interaction of external photons with free electrons on the metal surface., thus satisfying Maxwell equations.

As shown in Figure 1, on a flat semi-infinite metal surface, the interface between the metal and the medium is assumed to be on the yx plane, z = 0, and the normal direction is the z-axis. The magnetic field of the incident light is incident on the interface along the y-axis direction; the surface plasmon propagates along the x-axis direction. The region of z < 0 shows a metal, the dielectric constant is $\varepsilon_1$, and the real part of the dielectric constant of the metal is a negative value ($Re(\varepsilon_1) < 0$). The region of z > 0 shows a vacuum or other dielectric material, and its dielectric constant is $\varepsilon_2$. The electric fields in metals and dielectrics (or vacuum) are $E_1$ and $E_2$, the magnetic fields are $H_1$ and $H_2$, the wave vectors of light are $k_1$ and $k_2$ respectively, and the wave vectors in a vacuum are $k_0$. The wave vector of the surface plasmon is $k_{spp}$. According to the Maxwell equation, the form of the electromagnetic field is set as follows:

in the region of z < 0,

$$H_1 = (0, H_{y1}, 0) \exp[i(k_{x1}x - k_{z1}z - \omega t)], \tag{1}$$

$$E_1 = (E_{x1}, 0, E_{z1}) \exp[i(k_{x1}x - k_{z1}z - \omega t)], \tag{2}$$

in the region of z > 0,

$$H_2 = (0, H_{y2}, 0) \exp[i(k_{x2}x + k_{z2}z - \omega t)], \tag{3}$$

$$E_2 = (E_{x2}, 0, E_{z2}) \exp[i(k_{x2}x + k_{z2}z - \omega t)]. \tag{4}$$

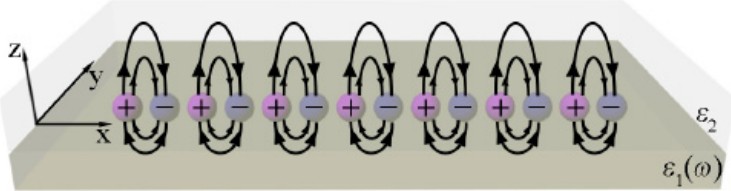

**Figure 1.** Surface plasmon polariton (SPP) propagation along an interface between metal and dielectric material.

According to the continuity of the electromagnetic field at the boundary z = 0, we obtain $E_{x1} = E_{x2}$, $H_{y1} = H_{y2}$, $k_{x1} = k_{x2}$. Let $k_{x1} = k_{x2} = k_{spp}$, and the electric field is determined by the Maxwell equation:

$$\nabla \times H = -ik_0 \varepsilon E, \tag{5}$$

thus, at x = 0,

$$\frac{k_{z1}}{\varepsilon_1} + \frac{k_{z2}}{\varepsilon_2} = 0. \tag{6}$$

In the above formula, $k_{z1}$ and $k_{z2}$ are positive numbers. Thus, only when the dielectric constant symbols of metals and dielectrics are opposite can surface plasmon waves be generated. The relationship between the wave vectors at the interface is $k_{xi}^2 + k_{zi}^2 = \varepsilon_i k_0^2$, which gives [50]

$$k_{spp} = k_0 \sqrt{\frac{\varepsilon_1 \varepsilon_2}{\varepsilon_1 + \varepsilon_2}}. \tag{7}$$

It is noted that $\varepsilon_1 < 0$, $|\varepsilon_2 + \varepsilon_1| < |\varepsilon_1|$; therefore, $k_{spp} > k_0$. In the dielectric material, $k_{z2}^2 < 0$, because $k_{z2}$ is an imaginary number. In the metal, $k_{z1}^2 = \varepsilon_1 k_0^2 - k_{spp}^2 < 0$, because $\varepsilon_1 < 0$. It can be inferred that the surface plasmons are exponentially weakened in both z and -z directions perpendicular to the metal surfaces, either through the medium (vacuum) or the metal, and can only propagate along the metal surface [51,52]. Due to the ohmic loss effect in the metal, the electric field strength decreases as the SPPs propagate along the metal surface. If a large propagation distance is desired, precious metals with small imaginary parts and large real parts, such as Ag and Au, should be used.

Using this characteristic of surface plasmons, some metal nanostructures can be fabricated, which can couple photons into surface plasmons and confine them to the metal surface at the nanometer scale, thus greatly compressing the spatial distribution scale of the electromagnetic field. The above theory provides a theoretical basis for the realization of nanolasers.

## 3. Structures and Characteristics of Plasmonic Nanolasers

### 3.1. Plasma Nanolaser Based on a Single Semiconductor Nanowire

This kind of laser uses nanowires to achieve the integration of the gain medium and the resonant cavity [53]. The photons generated by the nanowires are coupled with the metal layer to form surface plasmons, which propagate along the direction of the nanowire, transmit and oscillate in the F-P cavity formed by reflection at both ends of the nanowire, are amplified by the gain medium and achieve lasing. The working principle is shown in Figure 2.

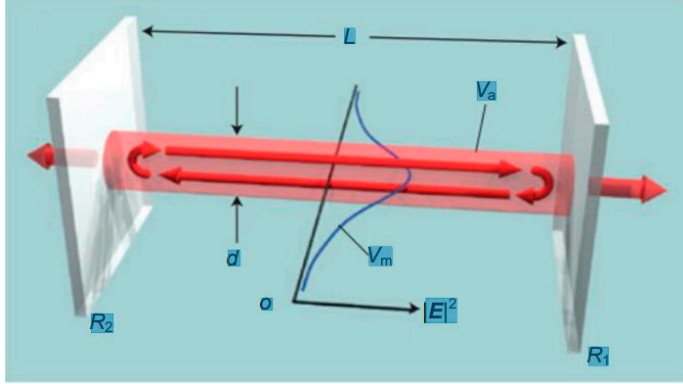

**Figure 2.** The working principle of the nanowire cavity [53].

In 2009, the Xiang Zhang group at the University of California, Berkeley reported a deep subwavelength hybrid plasma nanolaser based on nanowire structures [54]. Experiments have demonstrated that the laser can support a mode size as small as $\lambda^2/400$ due to the lasing behavior of the mixed mode formed by surface plasmons and the optical mode. The laser can confine light to a size that is less than 100 times the diffraction limit. As shown in Figure 3, the laser is composed of metal, medium and nanowire. The nanowire is placed on the metal, and the nanowire and the metal are separated by a nanometer-scale insulating dielectric layer. By using semiconductors and dielectric layers with large differences in their refractive indices, the optical field can be confined to a dielectric layer having a thickness of only a few nanometers. The composite surface plasmon mode supported by this structure has the advantages of a wide frequency band, small mode size, and low transmission loss. In this work, the nanowires used by the researchers were CdS semiconductor materials prepared by chemical vapor deposition. The characteristic wavelength was 489 nm, the dielectric layer was $MgF_2$, and the metal layer was silver with lower loss in the desired frequency range. Although the structure has the advantage of low propagation loss, the laser emission still operates at an ultra-low temperature (T < 10 K), and the pump optical power density is very high (about 100 $MW/cm^2$).

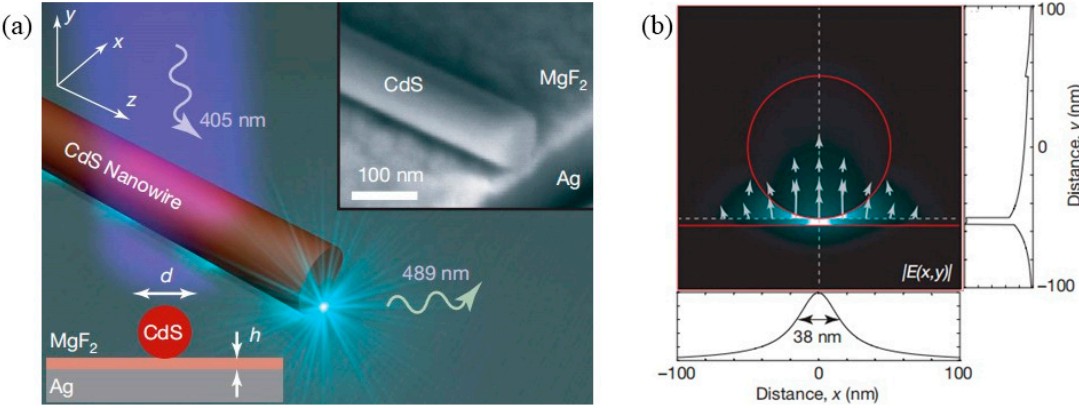

**Figure 3.** (**a**) Principle diagram and scanning electron microscope (SEM) picture of the hybrid plasmonic nanolaser. (**b**) Distribution of the model between the nanowire and the metal layer [54].

In the surface plasmon laser, the metallic material silver is the best choice for realizing a low-threshold surface plasmon laser because it has a small surface plasmon loss in the visible to near-infrared spectral range. However, because the silver films prepared by thermal evaporation or magnetron sputtering are usually polycrystalline and generally have a surface roughness of a few nanometers root mean square (RMS), the propagating loss of SPPs on the metal surface is so severe that the lasing threshold of surface plasmon nanolasers is still too high. This shortcoming restricts its practical application. Therefore, this kind of nanolaser is difficult to achieve lasing even under

continuous light pumping conditions. In 2012, Yu-Jung Lu et al. adopted an optimized two-step growth method to realize the epitaxy growth of silver thin films on (111)-type single crystal silicon [55]. A single crystal silver thin film with a thickness of 80 nm was obtained. A surface plasmon laser with a low threshold was obtained by continuous laser pumping at low temperature. Its structure is shown in Figure 4. Although this method can fabricate high-quality silver thin film, it has not been reported by other research groups that surface plasmon lasers are fabricated by similar processes, due to the complicated preparation process.

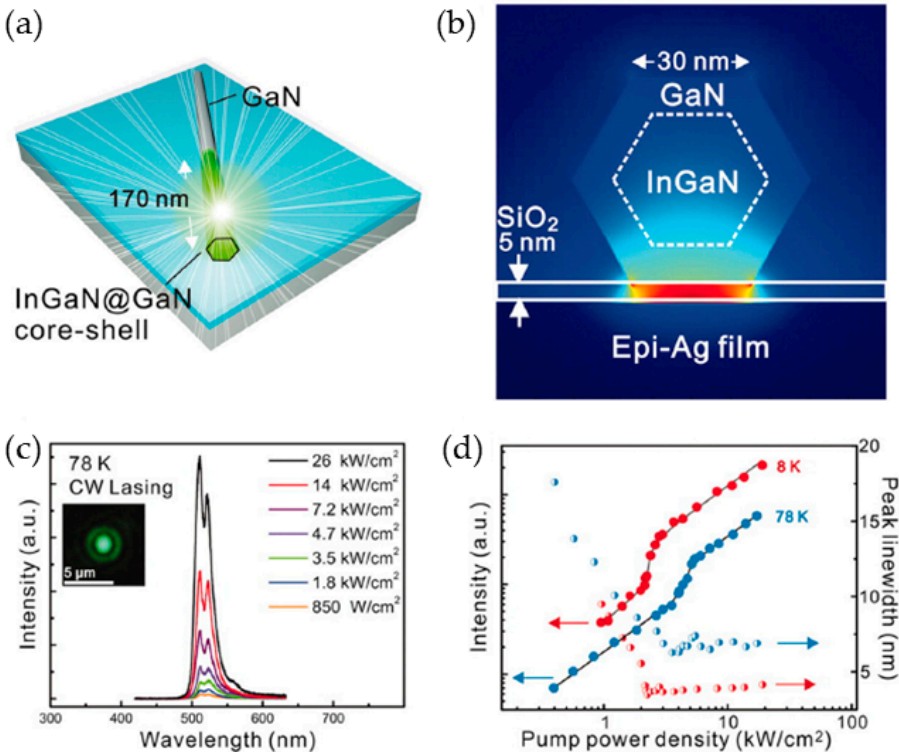

**Figure 4.** Plasmonic nanolaser using epitaxially grown silver film. (**a**) Schematic of the device, (**b**) calculated mode distribution, (**c**) lasing spectra, (**d**) L-L plots [55].

In this work, the surface plasmon resonator has the advantage of low loss, due to the silver layer having an atomically smooth surface, and the nanowires of the core–shell structure have good end faces and side morphology. The plasmonic nanolaser uses a continuous laser of 405 nm wavelength as the pump source, and the stimulated radiation spectrum has two lasing peaks at 510 nm and 520 nm. Its lasing threshold is only 2.1 kW/cm$^2$ and 3.7 kW/cm$^2$ at 8 K and 78 K temperatures.

The internal ohmic loss of metal will increase sharply when the optical wavelength is close to the energy range of the inter-band/in-band conversion of metal materials, thus reducing the propagation length of surface plasmons. Therefore, the realization of plasmonic nanolasers in the ultraviolet band is still a difficult task. In 2014, Qing Zhang et al. reported a surface plasmon laser with a lasing wavelength of 370 nm at room temperature, which is pumped by a laser with wavelength of 355 nm, a pulse width 10 ns and a repetition rate of 100 kHz [56]. The threshold is 35 mJcm$^{-2}$/3.5 MWcm$^{-2}$. The low-threshold characteristics of this laser are primarily due to high-quality single crystal nanowires and good semiconductor–medium–metal contact. As shown in Figure 5, the device forms a semiconductor–dielectric–metal structure by placing GaN nanowires on Al and separating Al from GaN by SiO$_2$. Al is used here because it has less loss in the ultraviolet band than silver. High quality GaN nanowires are the precondition of realizing low-threshold lasing. The nanowires are prepared by metal–organic chemical vapor deposition (MOCVD). The morphological characterization results show that the nanowires have a triangular cross section and smooth surface. The direct surface contact

between the nanowire and the substrate reduces the scattering loss, which is more advantageous for the exciton to transfer energy to the surface plasmon, meaning that the gain generated by the GaN nanowire is fully utilized.

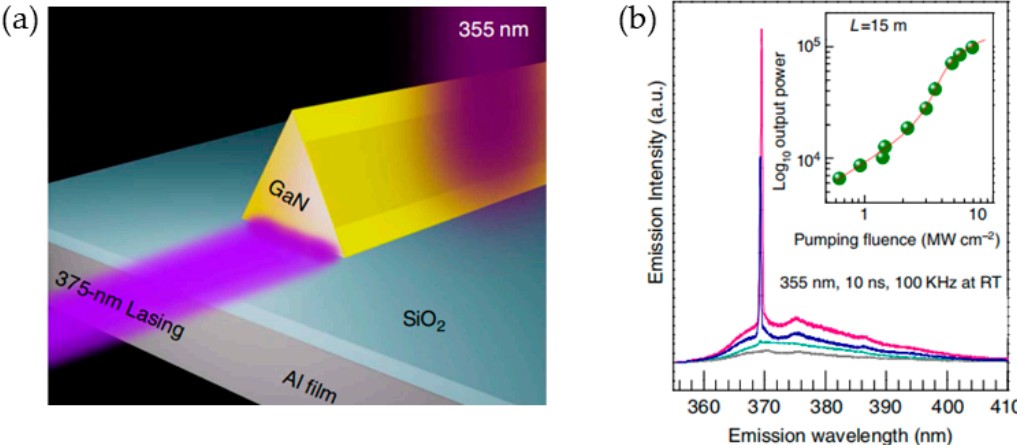

**Figure 5.** Room temperature ultraviolet plasmon laser. (**a**) Schematic of the device, (**b**) power-dependent spectra [56].

In 2016, Chou et al. obtained a low-threshold SPP laser by directly placing ZnO nanowires on the Al surface and optimizing the dielectric constant combination of the semiconductor and metal materials [57]. The device is shown in Figure 6. In their paper, a high-quality Al thin film with a single crystal structure were prepared by molecular beam epitaxy (MBE), which effectively reduced the additional scattering loss caused by polycrystalline Al thin films. The device is pumped by a laser source with wavelength of 355 nm, pulse width of 0.5 ns and a repetition rate of 1 kHz. At 77 K and 300 K temperatures, the threshold values of the laser are 16 MWcm$^{-2}$ and 110 MWcm$^{-2}$, respectively. The nanolaser can achieve 386 nm wavelength lasing at 353 K, which benefits from the single crystal Al film used in the device, the larger exciton oscillation intensity in ZnO and the simple laser structure.

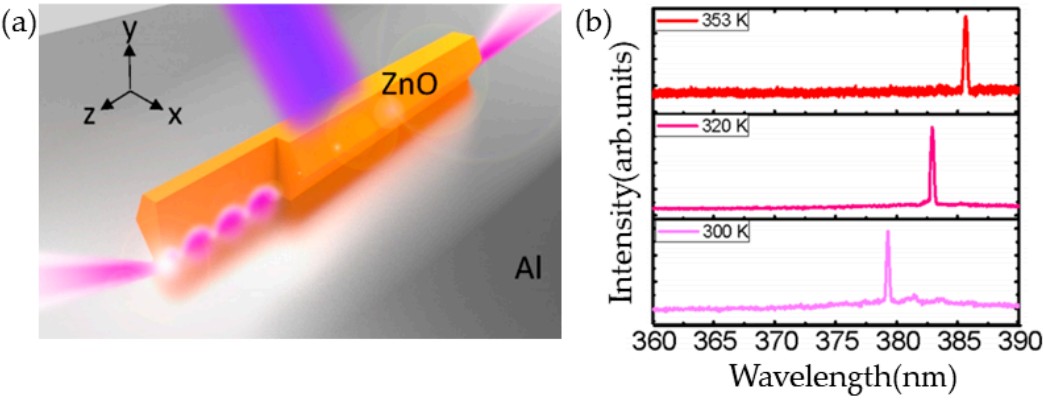

**Figure 6.** ZnO SPP nanolaser. (**a**) Schematic of the SPP nanolaser based on ZnO nanowires, (**b**) lasing spectra at different temperatures [57].

It can be seen from many studies that this kind of plasmonic nanolaser generally has a metal–dielectric–semiconductor structure. In order to reduce the loss caused by metal and the threshold of this nanolaser, three schemes have been proposed. First, in order to obtain high-quality gain dielectric nanowires with excellent morphology, the structural composition (such as the core–shell structure) and the preparation process of semiconductor nanowires should be improved. Second, the preparation process of the metal film should be improved; thereby, a high-quality metal film whose smooth surface is not easily oxidized can be obtained. Third, a suitable dielectric layer may be added between the

metal layer and the nanowire. The dielectric layer can store energy in the structure, thus reducing the energy loss between the interspace. However, under the coupled structure of the plasma mode and gain dielectric waveguide of semiconductor nanowires, there are still many problems in laser emission, such as the too-high gain threshold, low quantum efficiency, low luminescence efficiency and difficulty in beam quality control. The solution to these problems depends on the final solution of the resonant coupling mechanism of the semiconductor excitons with the plasma mode and the energy restriction mechanism of the plasma mode to resonant cavity mode.

### 3.2. Plasmonic Nanolaser Based on Single Perovskite Nanowires

In 2001, Huang et al. first studied the application of ZnO nanowires in deep-ultraviolet laser emission [58]. A series of subsequent studies show that material loss plays an important role in the threshold of plasmonic nanolasers. Although nanowires have many advantages as a gain media of laser emission, their application is limited by their high lasing threshold, so it is necessary to find more ideal optical gain materials. In recent years, metal halide perovskite materials with excellent optical conversion properties have attracted abundant attention. Metal halide perovskite material has a long carrier lifetime (10–100 ns) and a long carrier diffusion length (micron scale), and also has a high fluorescence quantum efficiency, which indicates that it has great potential as a gain medium.

In 2016, Eaton et al. of the University of California first reported the application of all-inorganic perovskite nanowires in a laser field [59]. They successfully prepared $CsPbBr_3$ and $CsPbCl_3$ nanowires by the liquid-phase cryogenic solution method and used them as gain media, as shown in Figure 7. Compared to the synthesis of semiconductor nanowires mentioned above, the preparation process and conditions of perovskite nanowires are much simpler. The nanowires prepared by the low-temperature solution method have a length of 2 to 40 µm and a width of 0.2 to 0.3 µm, which is an ideal gain medium. Under pulsed laser excitation, the lasing threshold of $CsPbBr_3$ nanowires is 5 µJcm$^{-2}$, and the maximum quality factor is about 1009. The time-resolved fluorescence spectroscopy results show that the radiative recombination lifetime of $CsPbBr_3$ nanowires is less than 30 ps under stimulated emission, which is much smaller than the radiation lifetime under normal conditions. The minimal recombination lifetime causes the carriers to deplete rapidly and be unable to participate in spontaneous emission, meaning that high-quality factor lasers can be obtained. Different from the double exciton recombination in the excitation process of semiconductor nanowires, the author attributes the excitation emission of all inorganic perovskite nanowires to the electron–hole plasma mechanism, which refers to two different gain mechanisms. Electron–hole plasma is formed when the carrier concentration exceeds the Mott density, which makes it possible to stimulate emission. The formation of the electron–hole plasma will reduce the refractive index and lead to the blue shift of the resonant mode. The experimental results provide evidence for the correctness of the electron–hole plasma mechanism. Compared with the previously reported organic–inorganic hybrid perovskite [60], all-inorganic perovskite exhibits better stability. Under the excitation of a pulsed laser, $CsPbBr_3$ nanowires can normally emit a laser for 109 cycles, equivalent to 1 h.

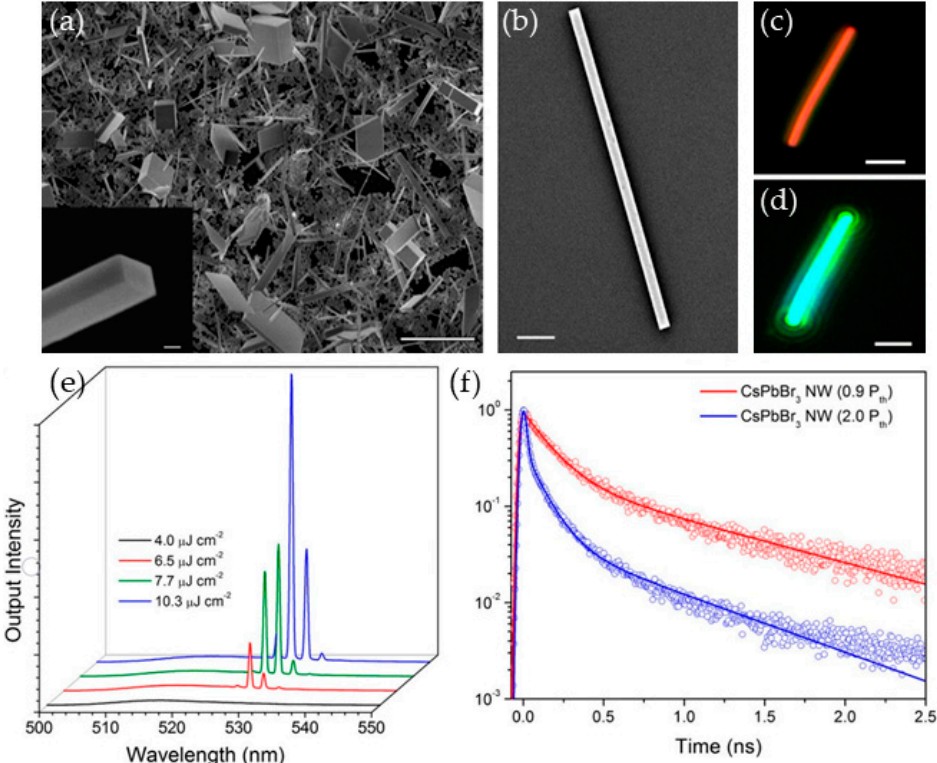

**Figure 7.** Applications of all-inorganic halide perovskite nanowires in laser field. (**a**) SEM (scanning electron microscope) images of CsPbBr$_3$ nanowires and nanoplates, (**b**) a single CsPbBr$_3$ nanowire isolated on a quartz substrate with a 5 nm Au sputter coat, (**c**) a dark-field photo of a single CsPbBr$_3$ nanowire below the lasing threshold, (**d**) a dark-field photo of a single CsPbBr$_3$ nanowire above the lasing threshold, (**e**) the power-dependent emission spectra of a single CsPbBr$_3$ nanowire, and (**f**) the time-resolved fluorescent spectra of CsPbBr$_3$ nanowires below (red) and above (blue) the lasing threshold [59].

### 3.3. Nanowire Array Plasmonic Nanolaser

In 2017, Huang et al. from the University of Electronic Science and Technology of China reported the design and manufacture of a new type of regular array plasmonic nanolaser that works at room temperature [61].

The structure is shown in Figure 8a. In this work, firstly, n-type ZnO nanowires (NWs) were fabricated on p-type GaN substrates by using chemical vapor deposition (CVD) and other subsequent microfabrication processes. The regular NW array has a length of about 5.1 μm and a diameter of 120 ± 10 nm. In order to obtain the regular array growth of NWs on the substrate, the growth positions of NWs are controlled in the regular hole array on the substrate, which is fabricated by electron beam lithography (EBL). Ag–dielectric hybrid nanofilms were sputtered on ZnO/GaN nanojunction arrays to excite surface plasmas at the interface of NWs and to ensure the limitation of plasma mode on nanolasers. Then, the laser behavior of the nano-heterojunction array device is experimentally demonstrated by the optical pumping method. The optical pumping threshold of the plasmonic nanolaser is about 184.9 mW/cm$^2$. As shown in Figure 8d, all the peak modes are at the same wavelength (387.5 nm) at different pump power densities; that is to say, the excitation light comes from the same mode. However, due to the high pump power and the excitation of an adjacent NW which is slightly different in length, another mode begins to appear. Based on the highest peak mode of the measured spectrum, the quality factor of the fabricated device is 1937.5. Compared with other nanolasers confined by the mode of surface plasmon, it has a significant quality factor advantage. Laser beam point mapping experiments show that with the increase of pump power density, as shown

in Figure 8e–f, the laser intensity gradually increases, the light gradually concentrates, and the laser beam is combined in an angle range of +30 degrees.

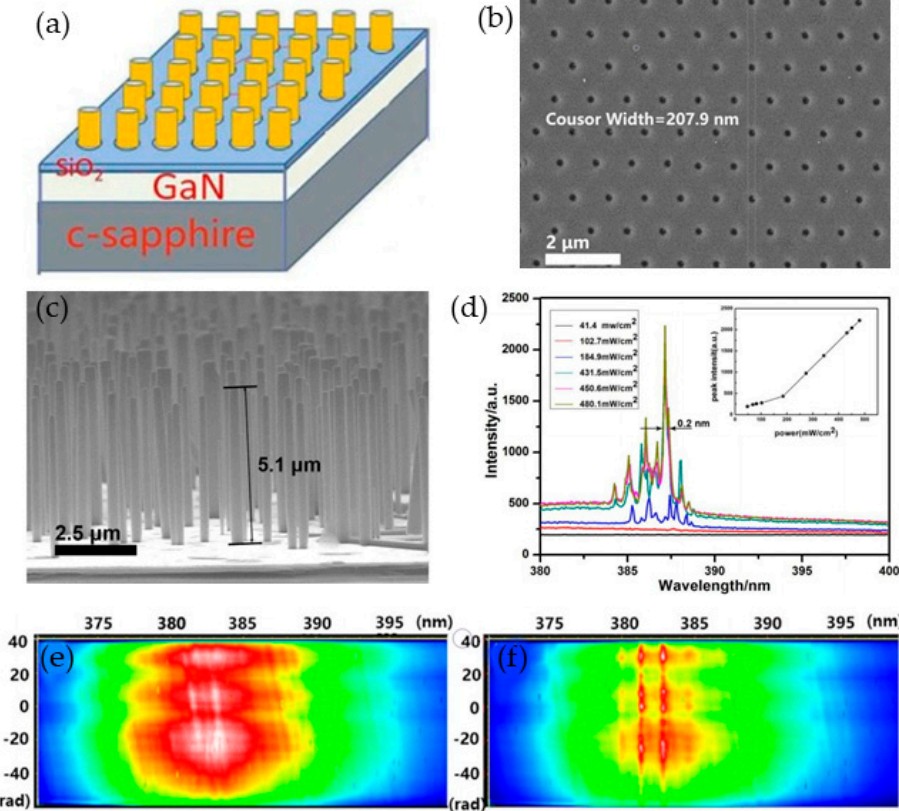

**Figure 8.** (**a**) Schematic of the novel regularly arrayed plasmonic nanolasers, (**b**) a SEM image of a regular hexagonal array of nanoholes on the substrate, (**c**) side view SEM image of an n-type vertically aligned ZnO nanowire (NW) (the length is about 5.1 μm) array, (**d**) optically pumped lasing spectra of the device at room temperature, and (**e**–**f**) beam spot mapping images at pump power densities of 136.9 and 205.4 mW/cm$^2$, respectively [61].

### 3.4. LSP-Based Nanolaser

LSPs are non-propagating surface plasmon excitations generated by conducting electrons in metal nanostructures coupled with electromagnetic fields. The LSP-based nanolaser is composed by replacing the medium around the metal particles with material with gain characteristics. The metal particles themselves are the resonator of the nanolaser. Compared with SPPS-based nanolasers, LSP-based nanolasers have received less attention.

In 2009, Mikhail A. Noginov of Norfolk State University first successfully demonstrated LSP-based nano-lasers [62]. As shown in Figure 9, the nanolaser consists of an Au core, providing the plasmon mode, and silicon dioxide doped with OG-488 dye molecule providing gain. The diameter of the Au core is 14 nm, the thickness of the silicon dioxide layer is 15 nm, and the diameter of the whole device is only 44 nm. Because this single nanolaser has a small scale relative to the lasing wavelength, the radiation loss is very small. The loss of LSPs depends almost entirely on the absorption effect. It is pointed out that the laser characteristics are observed under the condition of high pump power. The lasing spectral bands of the plasma laser are 480 ± 5 nm and 520 ± 20 nm, respectively, at the pump power density of 0.98 MWcm$^{-2}$ and 0.042 MWcm$^{-2}$.

In 2013, Meng et al. proposed a tunable wavelength surface plasmon nanolaser based on LSPs, in which an organic dye molecule is used as a gain medium [63]. Its structure is shown in Figure 10. The fabrication process of the device is as follows: firstly, Au/SiO$_2$ nanorods are synthesized, then the monolayer nanorods are dispersed on the surface of SiO$_2$ substrate, and the device is finally coated

with polyvinyl alcohol doped with R6G dye molecules. The Au nanorods are prepared by a crystal seed process. Sodium hydroxide solution and tetraethyl orthosilicate dissolved in methanol are poured into an aqueous solution containing Au nanorods. Then, an SiO₂ shell encapsulating Au is formed by the reaction. Unlike the spherical Au nanoparticles used by Mikhail A. Noginov et al., the nanorods used in this work can support both horizontal and longitudinal surface plasmon resonance modes. The simulation results show that the longitudinal mode can support better local field enhancement and thus provide stronger optical feedback. The device is pumped by a frequency-doubled laser with a center wavelength of 532 nm, a repetition rate of 1 Hz, and a pulse width of 25 ps. Under this pumping condition, the threshold of the laser reaches the micro-joule level. By adjusting the doping concentration of the dye, the tuning range of the laser wavelength from 562 nm to 627 nm is achieved.

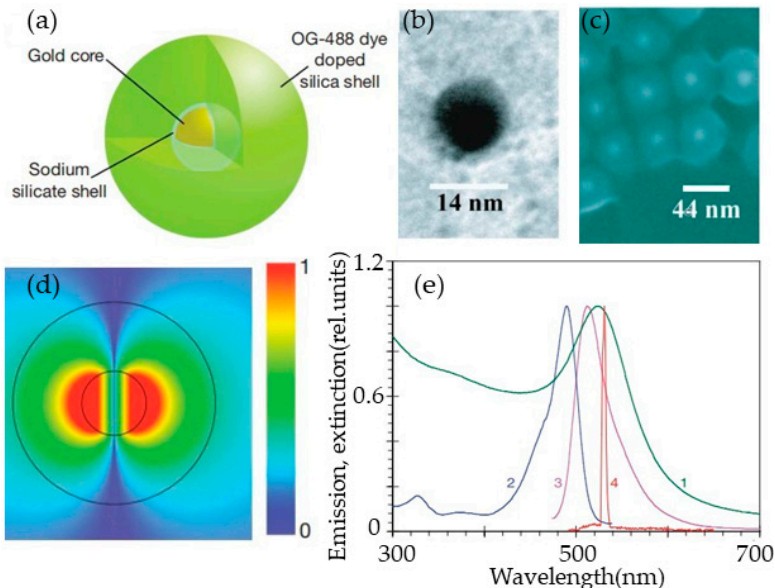

**Figure 9.** (**a**) Structure of the localized surface plasmon (LSP) nanolaser, (**b**) transmission electron microscope image of Au core, (**c**) scanning electron microscope image, (**d**) lasing mode with λ = 525 nm and Q = 14.8, and (**e**) spectroscopic results [62].

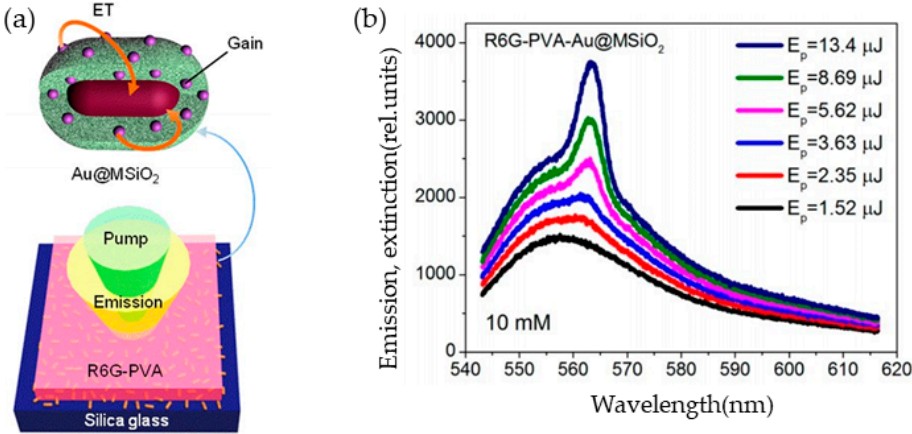

**Figure 10.** (**a**) Schematic of the LSP nanolaser, (**b**) emission spectra [63].

### 3.5. Nanolaser Based on Whisper-Gallery Effect

The whisper-gallery (W-G) effect is often applied to photonic crystal microcavities and disc-shaped microcavity nanolasers [64–67]. In recent years, this effect has also been used to match the surface plasmons provided by metals to fabricate nanoscale laser devices.

In 2011, Zhang et al. of the University of California at Berkeley first demonstrated experimentally that a semiconductor surface plasmon nanolaser based on the W-G effect can be realized at room temperature, which can compress light to $1/20^{th}$ of the wavelength at a vertical scale [68]. The preparation method of this plasmonic nanolaser is as follows: firstly, the CdS nanobelt is prepared by chemical vapor deposition (CVD). Then, the CdS nanobelt is cleaned by an ultrasonic wave in the solution matrix. Finally, 5 nm thick $MgF_2$ and 300 nm thick silver are deposited on the nanobelt. A Ti sapphire laser is used to pump the laser with a wavelength of 405 nm, a repetition rate of 10 kHz and a pulse width of 100 fs. A 20x objective lens is used to focus the pump source on a spot with a diameter of only 5 μm. The entire experiment was operated at room temperature.

Figure 11 shows the surface plasmon laser at room temperature and its SEM. The $MgF_2$ layer can not only reduce the loss of metal, but also provides mode limitation beyond the diffraction. The refractive index difference between CdS and $MgF_2$ makes it possible for the structure to confine the optical mode mainly to the $MgF_2$ layer, which provides a good mode limitation. At the same time, the metal absorption loss is greatly reduced by coupling the electric field from the metal surface to the $MgF_2$ layer. This coupling is very strong, which makes the momentum of the optical modes in the square CdS cavity increase greatly. That is, the effective refractive index increases greatly, meaning that the TM mode can obtain a sufficient internal reflection. For lasing light with wavelengths of 495.5 and 508.4 nm, the Q values are 97 and 38, respectively. The optical pump power densities of the threshold are 3070 and 4020 $MW/cm^2$, respectively. In this design, the shortcomings of the square structure are obvious. Because of the high symmetry, there are more modes, and so there are many lasing peaks, as shown in Figure 11c.

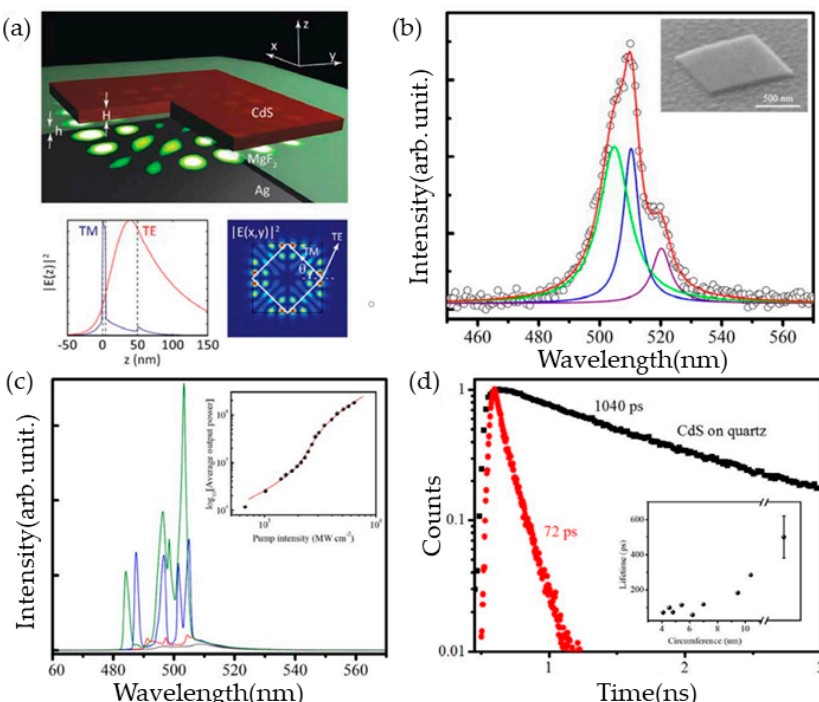

**Figure 11.** (**a**) Schematic of the room temperature plasmon laser. Bottom-left: the electric field distribution of TM and TE modes in the z direction. Bottom-right: the electromagnetic energy distribution of a TM mode in the x and y directions. (**b**) Spectrum and de-convolved modes at room temperature and low pump intensities (290 MW cm$^{-2}$) and SEM micrograph. (**c**) Optically pumped lasing spectra of the device at room temperature. (**d**) Time-resolved emission under weak pumping of the plasmon laser and CdS on quartz [68].

## 4. Existing Problems and Development Trends

It can be seen that the plasmonic nanolaser has great advantages in terms of its size and luminous intensity, but it is currently in the initial stage of research, and so it has many defects. There are still

many problems globally in the basic principle, preparation technology, pumping luminescence test and heat dissipation of plasma nanolasers which need to be solved urgently. Firstly, metal is introduced into the plasma nanolaser structure to form surface plasmons, so there is a high loss, which greatly affects the laser emission. The wrinkles and grooves on the metal surface will greatly aggravate the loss of plasma polaritons. Secondly, there are difficulties in the way of pumping. For example, it is difficult to fabricate nanoscale micro-electrodes if using electric pumping, and the loss will increase at room temperature. Finally, in the case of obtaining a lower loss and threshold, making nanolasers smaller and confining the light field of nanolasers to a smaller range is a research hotspot and difficult problem at the present stage. Sun et al. uniquely proposed the three-dimensional optical field limitation based on the principle of cosmo-temporal symmetry [69]. Compared with the two-dimensional plane, it greatly enhanced the optical field limitation ability of plasma nanolasers, but this is only a theoretical and simulation study.

With the new-type and miniaturization requirements of plasmonic nanolasers, and the development of integrated plasmonic nanolasers, it can be seen that the development trend of plasmonic nanolasers is mainly reflected in the following:

1) Developing from a single operating wavelength to a range of coordinated wavelengths;
2) Changing pumping methods from optical pumping to electric pumping;
3) Developing the gain medium of the resonator from cylindrical to polygonal core-shell composite nanowires in structure, from single nanowires to regular array nanowires in quantity, and from traditional semiconductors to new materials such as organic dye molecules and perovskite;
4) Developing the type of resonator from the F-P cavity composed of nanowires to the metal nanoparticle cavity to the echo wall cavity based on the W-G effect;
5) Increasing the working temperature from ultra-low temperature (T < 10K) to room temperature and developing towards higher temperatures;
6) Developing from an independent monomer structure to an integrated array.

## 5. Conclusions

In summary, surface plasmon nanolasers are currently a research hotspot globally due to their broad application prospects. The working principle, structure, characteristics and applications of plasmonic nanolasers, as well as their existing problems, are summarized. It can be predicted that in the next few years, various new structures of surface plasmon nanolasers will come out. Although the working conditions and volume requirements are different according to different application directions, a move towards a more miniaturized direction on the whole is inevitable. On this basis, minimizing the metal loss, achieving a working range above room temperature, and achieving photon and electronic technology in nano-scale collaborative work will be the focus of further research into nanolasers. Apply surface plasmon nanolasers to biomedical, chemical, nanolithography, optical interconnections of information transmission, data storage and other aspects will be another hotspot and focus of technical research in the future.

**Author Contributions:** F.L., S.L. and L.X. conceived the idea; F.L., L.X. and F.Y. wrote the paper; F.L., Y.L. and L.X. advised the paper. All authors reviewed the paper.

**Funding:** This research was funded by the National Natural Science Foundation of China (Grant No. 11204222), the Natural Science Foundation of Hubei Province, China (Grant No. 2013CFB316, Grant No. 2014CFB793), the Innovation Fund of School of Science, Wuhan Institute of Technology (No. CX2016106).

**Acknowledgments:** The authors would like to acknowledge the support of the Hubei Key Laboratory of Optical Information and Pattern Recognition, Wuhan Institute of Technology.

**Conflicts of Interest:** The authors declare no conflict of interest.

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
