# Peer review of "Surface Plasmon Nanolaser: Principle, Structure, Characteristics and Applications"

_applsci, doi:10.3390/app9050861_

Reviewer 1 Report

In this manuscript the authors introduced the plasmonic nanolasers that have been experimentally proven and made breakthroughs in recent years. In particular, the plasmonic nanolaser based on metal halide perovskites nanowires that replace conventional semiconductor nanowires have been highlighted. 

In this manuscript the authors introduced the plasmonic nanolasers that have been experimentally proven and made breakthroughs in recent years. In particular, the plasmonic nanolaser based on metal halide perovskites nanowires that replace conventional semiconductor nanowires have been highlighted.

To my view, this is a good work and the paper is written very well and clear, but the only issue realted to the first part of the paper and refreences. other parts are quite fine and very clear.

it would be really great if they can make the intorduction better and highlight previouce achivment from other groups.

in Ref:

please also cite this work too:

(1) https://pubs.rsc.org/en/content/articlelanding/2016/nr/c5nr08979d/unauth#!divAbstract

(2) 

https://www.osapublishing.org/ol/abstract.cfm?uri=ol-42-3-486

(3) 

https://www.osapublishing.org/ol/abstract.cfm?uri=ol-39-2-189

I can support this work for publication only if the authors can fallow my suggestions and my commetns.

Author Response

1.To my view, this is a good work and the paper is written very well and clear, but the only issue realted to the first part of the paper and refreences. other parts are quite fine and very clear.

it would be really great if they can make the intorduction better and highlight previouce achivment from other groups.

in Ref:

please also cite this work too:

(1)https://pubs.rsc.org/en/content/articlelanding/2016/nr/c5nr08979d/unauth#!divAbstract

(2) https://www.osapublishing.org/ol/abstract.cfm?uri=ol-42-3-486

(3) https://www.osapublishing.org/ol/abstract.cfm?uri=ol-39-2-189

Answers: Thank you for your advice. We have revised the introduction and quoted the above literature. The modifications have been marked in yellow.

[65] Abbas Madani, Libo Ma, Shading Miao, Matthew R. Jorgensen, Oliver G. Schmidt. Luminescent nanoparticles embedded in TiO2 microtube cavities for the activation of whispering-gallery-modes extending from the visible to the near infrared. Nanoscale 2016, 8, 9498-9503.

[66] Abbas Madani, Stefan M. Harazim, Vladimir A. Bolaños Quiñones, Moritz Kleinert, Andreas Finn, Ehsan Saei Ghareh Naz, Libo Ma, and Oliver G. Schmidt. Optical microtube cavities monolithically integrated on photonic chips for optofluidic sensing. Optics Letters 2017, 42, 486-489.

Reviewer 2 Report

The paper is a review of surface plasmon nano lasers, from principle to applications.  It is honestly written, but I have several general comments, that cannot allow me to consider the manuscript suitable for publication as it is. In my opinion, the introduction is quite poor and should be improved. For example, it should be proper to emphasize possible real applications of the devices in the future. I suggest to stress this point. 

Concerning the section ‘Basic principle of surface plasmons’, maybe it can be slightly compressed and adapted to a paper. 

Strictly considering the technical parts, there is no mention (I did not find it)  in the text about coherence properties of those lasers. Can you add some details about that?

Lines 9-16 The ‘featured applications’ section is not well focused. It sounds as an abstract.

Lines 31-34 The first 2 sentences sound rhetorical

Lines 139-141 you wrote the same sentence twice.

Lines 155-156 several RMS for nanometer surface ... what does it mean?

Are Figures 2-8 courtesy of authors cited in the paper?

Author Response

1. The paper is a review of surface plasmon nano lasers, from principle to applications.  It is honestly written, but I have several general comments, that cannot allow me to consider the manuscript suitable for publication as it is. In my opinion, the introduction is quite poor and should be improved. For example, it should be proper to emphasize possible real applications of the devices in the future. I suggest to stress this point. 

Answers: Thank you for your advice. We have revised the introduction according to your opinion. The modifications have been marked in yellow.

2. Concerning the section ‘Basic principle of surface plasmons’, maybe it can be slightly compressed and adapted to a paper. 

Answers: We have compressed this part.

3. Strictly considering the technical parts, there is no mention (I did not find it)  in the text about coherence properties of those lasers. Can you add some details about that?

Answers: In the examples reviewed in this paper, the preparation process of each nanolaser, specific volume, operating temperature, pumping conditions, lasing peak and threshold are given. Due to the length of the article, some of the examples only refer to the methods used in the preparation process and are not specifically described. It can be viewed in the reference.

4. Lines 9-16 The ‘featured applications’ section is not well focused. It sounds as an abstract. 

Answers: The ‘featured applications’ section has been modified.

5. Lines 31-34 The first 2 sentences sound rhetorical

Answers: This is what an expert said when I attended an academic forum. Laser processing has greatly promoted the development of productivity. This is a fact.

6. Lines 139-141 you wrote the same sentence twice. 

Answers: we have deleted the same sentence.(line132-133)

7. Lines 155-156 several RMS for nanometer surface ... what does it mean?

Answers: We have re-described the sentence. (line145-147) Surface roughness often shortened to roughness, is a component of surface texture. It is quantified by the deviations in the direction of the normal vector.

8. Are Figures 2-8 courtesy of authors cited in the paper?

Answers: The figures quoted in the article have been approved by the original published journal.

Round  2

Reviewer 1 Report

I can accept this work for publication

Reviewer 2 Report

Dear Authors, 

you exhaustively clarified my concerns.

I think the paper is now ready for publication.